# Association of Physcion and Chitosan Can Efficiently Control Powdery Mildew in *Rosa roxburghii*

**DOI:** 10.3390/antibiotics11111661

**Published:** 2022-11-19

**Authors:** Cheng Zhang, Jiaohong Li, Yue Su, Xiaomao Wu

**Affiliations:** 1School of Public Health, Guizhou Medical University, Guiyang 550025, China; 2Institute of Crop Protection, College of Agriculture, Guizhou University, Guiyang 550025, China; 3Department of Food and Medicine, Guizhou Vocational College of Agriculture, Qingzhen 551400, China

**Keywords:** powdery mildew, physcion, chitosan, *Rosa roxburghii*, natural products

## Abstract

Powdery mildew is an extremely serious disease of all *Rosa roxburghii* production regions in China and frequently causes 30~40% of economic losses. Natural products are considered excellent alternatives to chemical fungicides. In this work, we investigated the efficacy of physcion used together with chitosan controls *R. roxburghii* powdery mildew and impacts its resistance, growth, yield, and quality. The results reveal that the foliar application of 12.5 mg L^−1^ 0.5% physcion aqueous solutions (AS) + 250 mg L^−1^ chitosan efficiently controlled powdery mildew with the efficacies of 92.65% and 90.68% after 7 d and 14 d, respectively, which conspicuously (*p* < 0.05) higher than 83.62% and 80.43% of 25 mg L^−1^ 0.5% physcion AS, as well as 70.75% and 77.80% of 500 mg L^−1^ chitosan. Meanwhile, this association prominently ameliorated the resistant and photosynthetic capabilities of *R. roxburghii*. Simultaneously, this association was more efficient than physcion or chitosan alone for ameliorating the yield and quality of *R. roxburghii*. This work emphasizes that the association of physcion and chitosan can be nominated as a natural, efficient and environmental-friendly alternative ingredient in controlling *R. roxburghii* powdery mildew and ameliorating its resistant, photosynthesis, yield, and quality.

## 1. Introduction

*Rosa roxburghii* Tratt., a medical, edible, and ornamental perennial plant, is mainly produced in Guizhou, Yunnan, Sichuan, Hubei, Chongqing, Shaanxi, and Hunan provinces in China [1,2]. Its fruits are rich in ascorbic acid, superoxide dismutase (SOD), flavonoids, amino acids, and mineral elements, etc., and have been widely used for ameliorating immunity, improving digestive system, reducing blood pressure, and curing oral and gastric ulcers [1,2,3,4]. Many pharmacological studies have suggested that *R. roxburghii* fruits also possess anti-tumor, anti-oxidation, anti-atherosclerosis, anti-radiation, and anti-aging functions [4,5,6]. Currently, *R. roxburghii* has been widely grown as an economic fruit in Guizhou Province with a cultivation area of more than 170,000 hm^2^, and has become the biggest producing base in the world [7,8]. Powdery mildew caused by *Sphaerotheca pannosa* (Wallr.) Lev is an extremely serious fungal disease of all *R. roxburghii* production regions in China, which has been responsible for serious growth restrictions and quality declines, as well as 30~40% of yield or economic losses [7,8,9]. Although several chemical fungicides (e.g., pyraclostrobin, prothioconazole, myclobutanil, and tebuconazole, etc.) have been reported to control powdery mildew by local research [10,11]. Nevertheless, chemical fungicide residuals will inevitably pose a potential threat to the ecosystem and human beings, and also result in a buildup in tolerance in pathogens to fungicides with the increase of the use frequency [12,13,14]. In consideration of the serious perniciousness of powdery mildew and the reduced use of chemical fungicides, it is therefore imperative to quest for diversified alternative and environmental-friendly control measures against *R. roxburghii* powdery mildew.

Biological control of powdery mildew using basically harmless natural products is considered a feasible alternative to chemical fungicides [15,16,17]. Natural products are growingly favoured by the public and are also more and more applied in agriculture production [16,17]. For example, ascorbic acid could induce prevent and control *R. roxburghii* powdery mildew [18]. Moreover, our results in a previous study demonstrated that chitosan also induced control of *R. roxburghii* powdery mildew, and enhance its leaf photosynthesis and fruit quality [7]. Soon afterward, we further found that the mixed-use of chitosan and allicin viably controlled *R. roxburghii* powdery mildew with an effect of 85.97% [19]. In our recent report, we also manifested that chitosan could be used as a favorable assist to improve the effect of low-dose pyraclostrobin control *R. roxburghii* powdery mildew [9]. Many studies have also evidenced that chitosan is an inductor or biological fungicide for preventing and controlling plant’s diseases, as well as also a growth enhancer for boosting plant’s growth [20,21,22,23,24,25]. In that case, whether chitosan can assist other natural products to more effectively control *R. roxburghii* powdery mildew of is worthy of further attention.

Physcion, a natural anthraquinone derivative, was isolated from the ethanol extract from roots of Chinese rhubarb (*Rheum officinale* Baill.), and has widely been applied to prevent and control the powdery mildew, gray mold, and downy mildew of various plants in China [26,27,28]. In agriculture, physcion has been evidenced to possess excellent biological activity against many plant pathogens, such as *Blumeria graminis*, *Podosphaera xanthii*, *Sphaerotheca fuliginea*, *Rhizoctonia solani*, *Botrytis cinerea*, *Pseudoperonospora cubensis*, and *Pyricularia grisea* [15,28,29,30,31,32]. Xiang et al. [33] further demonstrated that physcion increased the growth of tomato and enhanced its defense resistance to gray mold by boosting an endogenous plant defense response and working as a growth enhancer. Noteworthy, Zhang et al. [34] found that 12.5~25.0 mg L^−1^ 0.5% physcion AS had a significant inhibitory effect on *R. roxburghii* powdery mildew with the efficacy of 77.01~82.26% after 14 d of application. Meanwhile, Yang et al. [29] suggested that a synergistic interaction between physcion and chrysophanol against plant powdery mildew under the proportions ranging from 1:9 to 5:5 was observed. Furthermore, so far, there is no literature usable about the mixed use of physcion and chitosan controls *R. roxburghii* powdery mildew. Overall, whether the effect of physcion control *R. roxburghii* powdery mildew can be ameliorated by chitosan and its growth, yield, and quality can also be enhanced are worth further study.

In this work, the effect of the association of physcion and chitosan control *R. roxburghii* powdery mildew in the field was first researched. Subsequently, the impacts of this association on the resistant and photosynthetic capacities of *R. roxburghii* were assessed. Simultaneously, the impacts of this association on the weight, quality, and amino acids of *R. roxburghii* were determined. This study provides a natural, efficient and alternative measure for controlling *R. roxburghii* powdery mildew.

## 2. Materials and Methods

### 2.1. Natural Fungicide, Chemical, and Electrostatic Atomizer

0.5% Physcion AS was provided by Qingyuanbao Biotechnology Co., Ltd. (Bayannur, Inner Mongolia, China). Chitosan with a deacetylation of more than 90.00% was provided from Mingrui Bioengineering Co., Ltd. (Zhenzhou, Henan, China). Other chemicals were analytical or chromatographic grades. Electrostatic atomizer was provided by Qiming Machinery Co., Ltd. (Taizhou, Zhejiang, China).

### 2.2. R. roxburghii Orchard

The high-quality and wide-planted ‘Guinong 5’ *R. roxburghii* with a tree age of nine-year-old were used as the experimental targets. Field control experiment was conducted in an *R. roxburghii* orchard in Longli county, China (106°95′13′′ E, 26°54′36′′ N), and its *R. roxburghii* tree density was 106 plants per 666.7 m^2^. As well, the climatic and nutritional information of *R. roxburghii* garden is shown in Table 1.

### 2.3. Control Experiment

In our recent report, we found that 500 mg L^−1^ chitosan had a good control effect on *R. roxburghii* powdery mildew with an efficacy of 77.80% after 14 d of application [9]. Moreover, Zhang et al. [34] found that 12.5~25.0 mg L^−1^ 0.5% physcion AS had a significant inhibitory effect on *R. roxburghii* powdery mildew with the control efficacy of 77.01~82.26% after 14 d of application. Considering the possible synergistic interaction between chitosan and physcion, the reduced application of fungicides, and the economic cost of field control experiments, thus three natural fungicide groups and one control group were devised to control powdery mildew. They were: (1) 12.5 mg L^−1^ 0.5% physcion AS + 250 mg L^−1^ chitosan (Ph 12.5 + Ch 250), (2) 25 mg L^−1^ 0.5% physcion AS (Ph 150), (3) 500 mg L^−1^ chitosan (Ch 500), (4) irrigation water (as control). The foliar spray method was adopted for spraying fungicide liquid. Three replicates were set in each group, and twelve plots were completely random order. The diagonal five trees in each plot were adopted for evaluating. The above-ground part of *R. roxburghii* plants was sprayed with 1.50 L of fungicide liquid or clear water on April 5 and 12 in 2022, respectively. The disease index and control efficacy of powdery mildew were respectively surveyed on, before, and after spraying according to Zhang et al. [9]. The incidence levels were classified as: no incidence for 0 level, thin hyphae with 1~2 diseased lobules for 1 level, thick hyphae with 3~4 diseased lobules for 2 levels, dense hyphae with 5~6 diseased lobules for 3 levels, dense hyphae with exceeding 7 diseased lobules for 4 levels. The disease index, increased disease index value, and control efficacy were counted as Formulas (1)–(3), respectively.
Disease index = 100 × ∑ (Disease level value × Leaf number at each level)/(Total leaf number × the biggest level)(1)
Increased disease index value = Disease index before spraying − Disease index after spraying (2)
Control efficacy (%) = 100 × (1 − Increased disease index value of fungicide/Increased disease index value of control)(3)

### 2.4. Determination Methods

The photosynthetic rate (Pn), transpiration rate (Tr), and water use efficiency (WUE) of *R. roxburghii* leaves were surveyed by a portable LI-6400XT photosynthesis measurement (LI-COR Inc., Lincoln, NE, USA) on 12 May 2022 [35]. Subsequently, *R. roxburghii* leaves from five orientations on each tree were stochastically gathered for determining their chlorophyll content and disease resistance parameters. Chlorophyll content with an ethanol/acetone (*v*/*v*, 1:2) extraction was measured using a UV-5800PC spectrophotometer [9]. The method of Cao et al. [36] was used for determining the phenolics and flavonoids of leaves, 2.00 g of sample was ground in 20 mL of HCl-methyl alcohol (1%, *v*/*v*) and centrifuged (8000× *g*, 8 min, 4 °C) for extracting 1 h without light, then the supernatant was checked at OD 280 nm and OD 325 nm, respectively. Coomassie brilliant blue method was used for determining leaf protein content, 2.00 g of sample was added with 14 mL of distilled water, extracted by ultrasonic, centrifuged, and the result was measured by the mass concentration of bovine serum albumin [36]. Anthrone colourimetric method was used for determining the sugar content of leaves [36]. The Ninhydrin colourimetry method was used for determining the proline (Pro) content of leaves, 0.50 g of sample was added into 5 mL of 3% sulfosalicylic acid solution to extract in boiling water bath for 10 min, then cooled and filtered to prepare the extract solution; 2 mL of extraction solution was added into 2 mL of glacial acetic acid and 2 mL of acid ninhydrin reagent, heated in boiling water bath for 30 min, and the solution turned red. After cooling, it was added 4 mL of toluene to extract for about 20 s with a quick mixer, and centrifuged for 5 min (3000 r). The upper liquid was measured at OD 520 nm, and toluene as the control [36]. The thiobarbituric acid method was used for determining the malonaldehyde (MDA) content of leaves, 1.00 g of sample was added into 5 mL of 10% trichloroacetic acid ice bath to grind into homogenate, centrifuged for 10 min (4 °C, 4000 r), and the supernatant was used as the extract; 2 mL of extract was added into 2 mL of 0.6% thiobarbituric acid solution, boiled in a boiling water bath for 15 min, and cooled and centrifugated, and then measured at OD 450 nm and OD 532 nm, 2 mL of 10% trichloroacetic acid was used as the control [36]. The nitrogen blue tetrazole method was used for determining the SOD activity of leaves, 0.50 g of sample was added into 1 mL of 0.5 mol/L phosphoric acid buffer (pH = 7.8) and ground into homogenate in an ice bath, centrifuged for 20 min (4 °C, 4000 r); The 3 mL reaction system contained 0.3 mL of 750 umol/L azobenzene tetrazole solution, 0.05 mL of extract crude enzyme solution, 0.3 mL of 20 umol/L riboflavin solution, 0.3 mL of 130 mmol/L methionine solution, 0.3 mL of 100 umol/L EDTA-Na_2_, 1.5 mL of 0.05 mol/L phosphate buffer solution and 0.25 mL of distilled water; After mixing, one branch pipe was put in a dark place, and the other pipes reacted for 20 min under 4000 Lx light; After the reaction, the OD 560 nm was measured with a non-illuminated tube as the control, and one enzyme activity unit was 50% inhibition of NBT photoreduction [36]. The catechol method was used for determining the polyphenoloxidase (PPO) activity of leaves, 0.50 g of sample was added with 0.05 g of polyvinylpyrrolidone and 2 mL of 0.1 mol/L phosphate buffer (pH = 6.5) to grind into a homogenate in an ice bath; The volume was fixed to 5 mL, filtered with nylon cloth, centrifuged for 15 min (4 °C, 8000 r), and the supernatant was the crude enzyme extract; 0.2 mL of enzyme extract was added into 2.8 mL of phosphate buffer containing 0.02 mol/L catechol (0.1 mol/L, pH 6.8) to mix; After reacting in a water bath at 30 °C for 2 min, OD 398 nm was determined with the same volume of extract as the control, the change of OD 398 nm value of 0.01 per minute was taken as one enzyme activity unit [36].

*R. roxburghii* fruits from five orientations on each tree were stochastically harvested on 31 August 2022. The weight and yield of fruits were measured based on the method of Li et al. [7,19]. The ascorbic acid, sugar, solid, protein, acidity, flavonoids, triterpenes, and SOD activity of fruits were determined by the methods of Cao et al. [36]. High-performance liquid chromatography (HPLC, ThermoFisher U3000, Waltham, MA, USA) method was used for checking ascorbic acid, 0.25 g sample was added into 1.5 mL of 0.3% metaphosphoric acid solution, and then frozen, ground and mixed for 2 min, centrifuged at 4 °C for 15 min (5000 r), and the supernatant was put through 0.22 μm water phase filtration membrane to be measured; The chromatographic conditions were as follows: ZORBAX SB-C18 column (250 mm × 4.6 mm, 5 μm), 30 °C of column temperature was, diode array detector, 254 nm of the detection wavelength, 0.1 mol/L disodium phosphate solution (pH 2.7) of mobile phase, 1 mL/min of flow rate, 10 μL of injection volume. The soluble solid was determined by a digital refractometer (Deke Machinery Technology Co., Ltd., Hebei, China). The titratable acidity of fruits was measured by the NaOH titration method [36]. Triterpenes of fruits were measured by the vanillin-glacial acetic acid colourimetric spectrophotometry method, 1.00 g of sample was added into 50 mL absolute ethanol to ultrasonic extract for 1 h (50 °C), the supernatant was centrifuged, and then measured at OD 548 nm, its result was calculated by the mass concentration of ursolic acid [36]. The determination methods of sugar, protein, flavonoids, and SOD activity were the same as above. Additionally, hydrolyzed amino acids of fruits were measured by an HPLC system based on the method of Zhang et al. [37]. A total of 0.10 g of sample was added into a 20 mL hydrolysis tube and 16 mL of 6 mol/L HCl solution to vacuum degas for 30 min, filled nitrogen to seal the tube, hydrolyzed at 110 °C for 22~24 h and cooled down, and then transferred into a 50 mL volumetric flask for constant volume by the deionized water; 1 mL of hydrolysate was deacidified and drain under vacuum, and then added 1 mL of 0.02 mol/L HCl solution to fully dissolve it; 500 μL above solution was added 250 μL of 1 mol/L triethylamine acetonitrile solution and add 25 μL of 0.1 mol/L phenyl isothiocyanate acetonitrile solution, left at room temperature for 1 h, then added 2 mL n-hexane to shake it violently, and leaved it to stand for 10 min, finally, the lower solution was put through 0.22 μM aqueous phase membrane for analysis. Chromatographic conditions: 0.1 mol/L sodium acetate-acetonitrile solution (93:7, *v*/*v*) of mobile phase A, acetonitrile-water (8:2, *v*/*v*) of mobile phase B, 1.0 mL/min of flow rate, 10 μL of injection volume, 254 nm of wavelength, and 40 °C of column temperature, as well as the column filler, was octadecyl silane bonded silica gel (4.6 mm × 250 mm, 5 μm). Subsequently, essential, nonessential, and total amino acids, as well as essential amino acids/nonessential amino acids and the proportion of essential amino acids in total amino acids were calculated.

### 2.5. Data Analyses

Data were expressed as the average value ± standard deviation (SD). The significant differences in data were checked by Duncan’s test with one-way analysis of variance (ANOVA) on an SPSS 18.0 system (SPSS Inc., Chicago, IL, USA). Origin 10.0 system (OriginLab, Northampton, MA, USA) was applied for editing figures.

## 3. Results

### 3.1. Efficacy of Physcion and Chitosan Control Powdery Mildew

Table 2 exhibits the efficacies of physcion + chitosan, physcion, and chitosan control *R. roxburghii* powdery mildew. Ph 12.5 + Ch 250, Ph 25, and Ch 500 significantly (*p* < 0.05) abated the disease index of powdery mildew after spraying fungicides. Ph 12.5 + Ch 250 showed a superlative control ability on powdery mildew with the efficacies of 90.68~92.65% after spraying fungicides, which observably (*p* < 0.05) greater than that of physcion or chitosan alone. Although Ch 500 exhibited an inferior control efficacy, its control efficacy gradually lasted after spraying and appeared a preferable persistence. Moreover, the amount of Ph 12.5 + Ch 250 effectively declined compared with Ph 25 or Ch 500. These results manifest that the association of physcion and chitosan efficiently alleviated the occurrence of powdery mildew, whose control efficacy was substantially (*p* < 0.05) better than that of physcion or chitosan alone.

### 3.2. Impacts of Physcion and Chitosan on Leaf Resistance

Figure 1 shows the impacts of physcion + chitosan, physcion, and chitosan on the resistant substances of *R. roxburghii* leaves. In comparison to the control, Ph 12.5 + Ch 250, Ph 25, and Ch 500 significantly (*p* < 0.05) heightened the flavonoids, protein, and sugar contents of leaves, and Ph 12.5 + Ch 250 and Ch 500 significantly (*p* < 0.05) promoted their phenolics. Meanwhile, the flavonoids, protein, and sugar of leaves controlled by Ph 12.5 + Ch 250 were markedly (*p* < 0.05) greater than those of Ph 25 or Ch 500. The phenolics of *R. roxburghii* leaves controlled by Ph 12.5 + Ch 250 were also observably (*p* < 0.05) greater than that of Ph 25, but was not significantly (*p* < 0.05) different from that of Ch 500. Moreover, the phenolics, flavonoids, protein, and sugar of leaves were no substantial (*p* < 0.05) differences in treatments of Ph 25 and Ch 500, and those of Ch 500 slightly exceeded those of Ph 25. The results suggest that physcion used together with chitosan notably boosted the phenolics, flavonoids, protein, and sugar of leaves compared with the alone application of physcion and chitosan, and thereby ameliorated *R. roxburghii* defence resistance to powdery mildew.

Figure 2 displays the impacts of physcion + chitosan, physcion, and chitosan on the Pro, MDA, and resistant enzyme activity of leaves. In comparison to the control, Ph 12.5 + Ch 250, Ph 25, and Ch 500 observably (*p* < 0.05) enhanced the Pro, SOD, and PPO of leaves and declined their MDA. Meanwhile, the Pro, SOD, and PPO of leaves controlled by Ph 25, and Ch 500 markedly (*p* < 0.05) less than those of Ph 12.5 + Ch 250. Leaf MDA controlled by Ph 12.5 + Ch 250 was substantially (*p* < 0.05) less than that of Ph 25, but was not markedly (*p* < 0.05) different from that of Ch 500. Moreover, the Pro, SOD, and PPO of leaves controlled by Ch 500 were higher than those of Ph 25, and MDA content was lower than that of Ph 25. These results further evidence that the association of physcion and chitosan efficiently ameliorated the Pro content and resistant enzyme activity of *R. roxburghii* leaves, downscaled their MDA, and soundly enhanced *R. roxburghii* stress resistance.

### 3.3. Impacts of Physcion and Chitosan on Leaf Photosynthesis

Figure 3 shows the impacts of physcion + chitosan, physcion and chitosan on the photosynthetic performance of leaves. In comparison to the control, Ph 12.5 + Ch 250, Ph 25, and Ch 500 observably (*p* < 0.05) augmented the chlorophyll content, Pn, and Tr of leaves, whereas their WUE had no substantial (*p* < 0.05) difference in all treatments. Concurrently, the chlorophyll, Pn, and Tr of leaves controlled by Ph 12.5 + Ch 250 were 6.36 μg g^−1^, 8.28 μmol CO_2_ m^−2^ s^−1^, and 2.95 mmol H_2_O m^−2^ s^−1^, which markedly (*p* < 0.05) greater than those of Ph 25, and Ch 500. Moreover, the chlorophyll content, Pn, and Tr of leaves controlled by Ch 500 were substantially (*p* < 0.05) greater than those of Ph 25. The results reveal that the association of physcion and chitosan could reliably improve *R. roxburghii* photosynthesis compared with the alone application of physcion and chitosan, and then further enhanced their flourishing growth.

### 3.4. Impacts of Physcion and Chitosan on Yield, Quality, and Amino Acids of Fruits

Figure 4 shows the impacts of physcion + chitosan, physcion, and chitosan on the weight and yield of fruits. Contrasted to the control, Ph 12.5 + Ch 250, Ph 25, and Ch 500 conspicuously (*p* < 0.05) promoted fruit weight and yield. The single fruit weight and fruit yield controlled by Ph 12.5 + Ch 250 were 20.78 g and 7.42 kg per plant, which markedly (*p* < 0.05) increased by 13.00%, 12.51%, and 33.72%, as well as 12.37%, 11.88%, and 32.97% compared to Ph 25, Ch 500, and control, respectively. Meanwhile, the single fruit weight and fruit yield had no substantial (*p* < 0.05) differences in Ph 25 and Ch 500. The results indicate that the association of physcion and chitosan efficiently stimulated the growth of fruits in contrast to physcion or chitosan alone. Table 3 depicts the impacts of physcion + chitosan, physcion, and chitosan on fruit quality. In comparison to the control, Ph 12.5 + Ch 250, Ph 25, and Ch 500 significantly (*p* < 0.05) increased the ascorbic acid, protein, sugar, acidity, solids, triterpenes, flavonoids, and SOD activity of fruits. Simultaneously, the ascorbic acid, protein, sugar, acidity, solids, triterpenes, flavonoids, and SOD activity of fruits controlled by Ph 12.5 + Ch 250 were significantly (*p* < 0.05) greater than those of Ph 25, and Ch 500. Furthermore, the aforementioned quality parameters of fruits had no substantial (*p* < 0.05) differences in Ph 25 and Ch 500. These findings suggest that the association of physcion and chitosan efficiently ameliorated the nutritional quality of fruits.

Table 4 shows the impacts of physcion + chitosan, physcion, and chitosan on fruit amino acids. In contrast to control, Ph 12.5 + Ch 250, Ph 25, and Ch 500 significantly (*p* < 0.05) upgraded fruit EAA, Ph 12.5 + Ch 250 significantly (*p* < 0.05) heighten its NAA, TAA, the proportion of EAA in TAA, as well as EAA/NAA. The EAA, TAA, or EAA/NAA of fruits controlled by Ph 12.5 + Ch 250 were higher than those of Ph 25. Meanwhile, the EAA, NAA, TAA, the proportion of EAA in TAA, or EAA/NAA of fruits managed by Ph 25 were slightly less than those of Ch 500. The results manifest that the boosting effect for *R. roxburghii* amino acids by physcion + chitosan was better than that of physcion or chitosan alone.

## 4. Discussion

Many types of research have evidenced that physcion could effectively prevent and control powdery mildew (*Blumeria graminis*, *Podosphaera xanthii*, *Sphaerotheca fuliginea,* etc.), grey mold (*Botrytis cinerea*), and downy mildew (*Pseudoperonospora cubensis*) of various plants [26,27,28,29,30,31,32,33]. Yang et al. [30] found that physcion could inhibit *B. graminis* of wheat powdery mildew by inhibiting conidia germination, increasing the appressorium deformation rate before pathogen infection, and declining haustoria length of and secondary haustoria number after infection. Ma et al. [32] reported that physcion induced the localised resistance of barley against powdery mildew, and Xiang et al. [33] further found that physcion could enhance the defence resistance of tomato to grey mold. For *R. roxburghii*, Zhang et al. [34] suggested that 12.5~25.0 mg L^−1^ 0.5% physcion AS could control its powdery mildew with an efficacy of 77.01~82.26%. Moreover, chitosan has a preferable antifungal activity and can induce plant resistance to various diseases [20,21,22,23,24,25]. In our previous study, our results indicated that chitosan notably induced control of *R. roxburghii* powdery mildew [7]. In this study, 12.5 mg L^−1^ 0.5% physcion AS + 250 mg L^−1^ chitosan could efficiently control *R. roxburghii* powdery mildew with the efficacies of 92.65% and 90.68% after 7 d and 14 d, respectively, which conspicuously (*p* < 0.05) greater than 83.62% and 80.43% of 25 mg L^−1^ 0.5% physcion AS, as well as 70.75% and 77.80% of 500 mg L^−1^ chitosan. The results show that the association of physcion and chitosan more efficiently controlled *R. roxburghii* powdery mildew compared to physcion or chitosan alone. Physcion + chitosan should have a conspicuous synergetic interaction: physcion prevented pathogen infection, killed pathogens, and induced defence resistance, while chitosan induced defence resistance and promoted growth.

The phenolics and flavonoids are considerable disease resistance compounds which can affect lignin biosynthesis and enhance host cell lignification [38]. Protein is closely related to plant’s defence resistance, and cell permeability can be regulated by Pro and soluble sugar, as well as MDA reflecting the peroxidation level of membrane lipid [38,39]. Moreover, SOD can obliterate free radicals in plants, while PPO is a key catalytic enzyme for the formation of lignin, phenolics and quinone, which are intimately linked with plant’s defence resistance [38]. Ma et al. [32] found that physcion controlled powdery mildew of barley by enhancing the expression of defence-related genes (especially leaf-specific thionin genes), and Xiang et al. [33] further reported that physcion could enhance tomato’s resistance to grey mold by stimulating an endogenous plant defence response. Many findings demonstrated that chitosan could enhance plants’ defence responses by increasing resistant compounds and stimulating defence enzyme activity [19,20,21,22,23,24,25,35,39,40,41]. Our previous studies have also manifested that chitosan could reliably raise the phenolics, flavonoids, protein, Pro, and sugar contents, as well as resistant enzyme activity of *R. roxburghii*, and decrease MDA content [7,9,19]. In this study, physcion used together with chitosan effectively boosted the phenolics, flavonoids, protein, Pro, sugar, SOD, and PPO of *R. roxburghii* compared with the alone application of physcion and chitosan, and thereby ameliorated its defense resistance to powdery mildew. Overall, physcion and chitosan had a noticeable synergetic interaction in ameliorating the defense resistance of *R. roxburghii* and hence controlling its powdery mildew.

Photosynthesis is the prerequisite of the growth, yield and quality of plants, and chlorophyll is an indispensable pigment in photosynthesis, as well as transpiration, is mainly driving force for the absorption and transport of water and nutrients in plant cells. Xiang et al. [33] indicated that physcion could act as a growth enhancer for significantly boosting tomato growth. Meanwhile, chitosan can enhance plant photosynthesis via raising chlorophyll, and thereby promote the growth and development of plants [21]. In previous results, we also found that chitosan, chitosan + allicin, or chitosan + pyraclostrobin could efficaciously promote the chlorophyll, Pn, and Tr of *R. roxburghii* [7,9,19]. In this work, the association of physcion and chitosan could reliably ameliorate the chlorophyll content, Pn, and Tr of *R. roxburghii* compared with the alone use of physcion and chitosan, and then further enhanced their flourishing growth. Additionally, chitosan can also act as a growth enhancer by activating the gene expression and signal transduction of cytokinin and auxin of plants [21,41]. The results here indicate that physcion used together with chitosan efficaciously stimulated the growth of *R. roxburghii* fruits contrasted to physcion or chitosan alone. These favourable effects might originate from the synergetic interaction between physcion and chitosan: they could not only protect *R. roxburghii* from pathogen infection, but also promote its healthy growth.

The fruit quality is seriously affected by powdery mildew and determined by the positive growth of *R. roxburghii*. The results in this study demonstrate that the vitamin C, protein, sugar, acidity, solid, flavonoids, triterpenes, and SOD activity of *R. roxburghii* fruits managed by the association of physcion and chitosan were notably (*p* < 0.05) greater than those of physcion or chitosan. It also shows that under the premise of no disease occurrence and healthy growth, the favourable nutritional quality of *R. roxburghii* was automatically formed. At the same time, the findings in this work manifest that the boosting efficacy for fruit amino acids by the association of physcion and chitosan was better than that of physcion or chitosan alone. According to the amino acid model offered by FAO, EAA/NAA and the proportion of EAA in TAA in the favorable foods are ≥0.6 and 40%, respectively [42]. In this work, EAA/NAA and the proportion of EAA in TAA of *R. roxburghii* managed by physcion + chitosan were 0.26 and 18.91% respectively, which were nearer to the ideal values than physcion or chitosan alone. The findings highlight that physcion and chitosan had an emphatic synergetic interaction in ameliorating *R. roxburghii* quality. 

Chemical fungicide residuals will inevitably pose a potential threat to the ecosystem and human beings [12,13,14]. Currently, natural products with non-toxicity, high efficacy and low risks as an effective alternative to chemical fungicides, are increasingly used in agriculture production and are also growingly preferred by consumers [15,16,17]. Physcion, a natural anthraquinone derivant, has been demonstrated to possess potential anti-neoplastic and anti-microbial characteristics and is widely used in pharmaceutical and agriculture fields [26,27,28,29,30,31,32,33,34,43,44]. Simultaneously, chitosan is a natural, non-toxic, biodegradable, and renewable product extensively used in food, cosmetics, and agriculture fields [22,23,24]. In this study, the association of physcion and chitosan could efficiently control *R. roxburghii* powdery mildew with the efficacy of 90.68~92.65%, as well as effectively ameliorate the resistance, photosynthesis, yield and quality of *R. roxburghii.* Moreover, 141 days (From 12 April to 31 August) of the safe interval time for fruits was immensely long. Consequently, the possible safety risks caused by physcion or chitosan are also practically nonexistent. The findings emphasize that 12.5 mg L^−1^ 0.5% physcion AS + 250 mg L^−1^ chitosan can be nominated as a candidate association for controlling *R. roxburghii* powdery mildew and ameliorating its resistance, growth, and quality. However, the interaction mechanism of physcion and chitosan on the pathogen of powdery mildew should be further revealed in future research, so as to fully understand the control mechanism of this association on *R. roxburghii* powdery mildew.

## 5. Conclusions

In conclusion, the association of physcion and chitosan more efficaciously controlled *R. roxburghi* powdery mildew compared with physcion or chitosan alone. Furthermore, the association of physcion and chitosan prominently ameliorated the defense resistance and photosynthetic capacity of leaves. Simultaneously, this association was also more efficacious than physcion or chitosan alone in enhancing the yield, quality and amino acids of fruits. This study emphasizes that the association of physcion and chitosan can be applied as a natural alternative ingredient for preventing and controlling powdery mildew in *R. roxburghii* and ameliorating its resistant, photosynthesis, yield and quality.

## Figures and Tables

**Figure 1 antibiotics-11-01661-f001:**
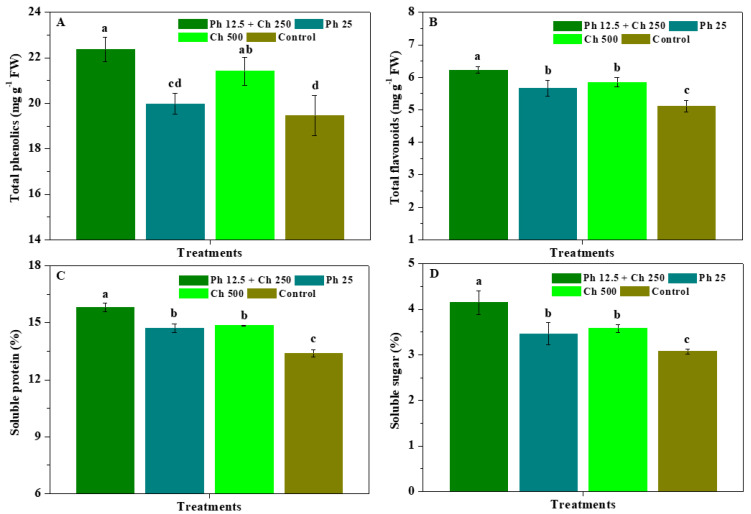
The impacts of physcion + chitosan, physcion, and chitosan on the phenolics (**A**), flavonoids (**B**), protein (**C**), and sugar (**D**) of leaves. The significant differences in the five treatments at 5% (*p* < 0.05) level are stood by different letters. Error bar represents the SD of three duplicates.

**Figure 2 antibiotics-11-01661-f002:**
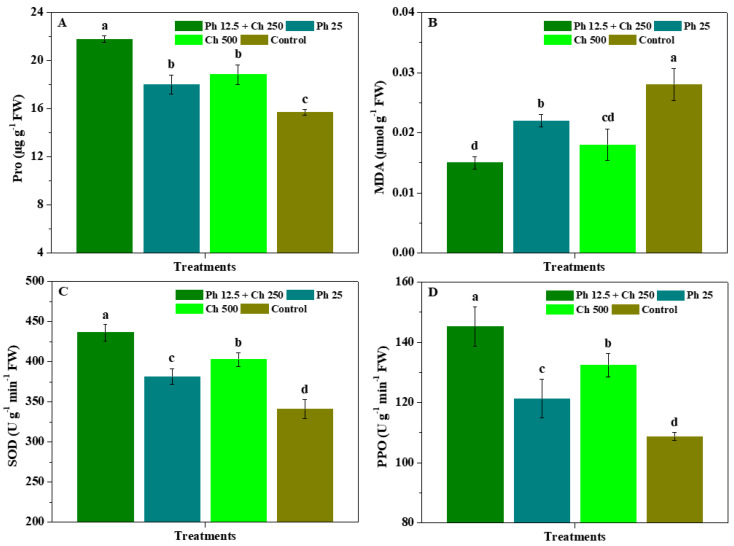
The impacts of physcion + chitosan, physcion, and chitosan on the Pro (**A**), MDA (**B**), SOD activity (**C**), and PPO activity (**D**) of leaves. The significant differences in the five treatments at 5% (*p* < 0.05) level are stood by different letters. Error bar represents the SD of three duplicates.

**Figure 3 antibiotics-11-01661-f003:**
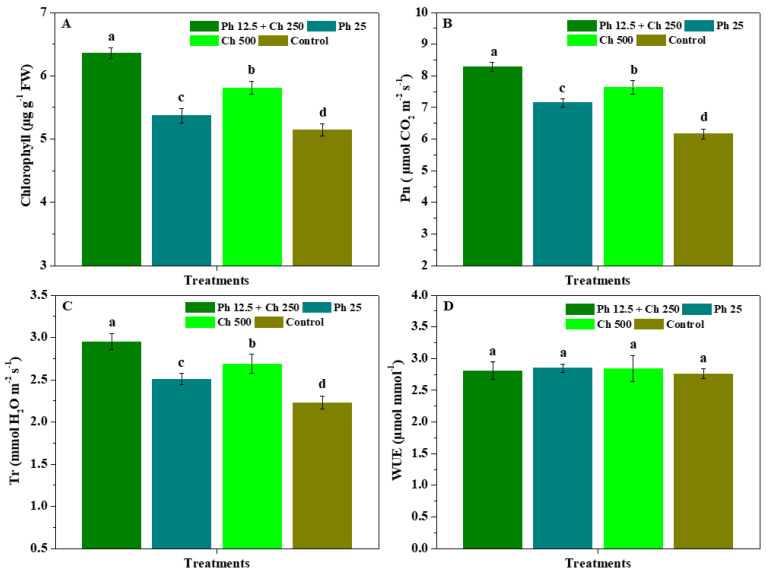
The impacts of physcion + chitosan, physcion, and chitosan on the chlorophyll (**A**), Pn (**B**), Tr (**C**) and WUE (**D**) of leaves. The significant differences in the five treatments at 5% (*p* < 0.05) level are stood by different letters. Error bar represents the SD of three duplicates.

**Figure 4 antibiotics-11-01661-f004:**
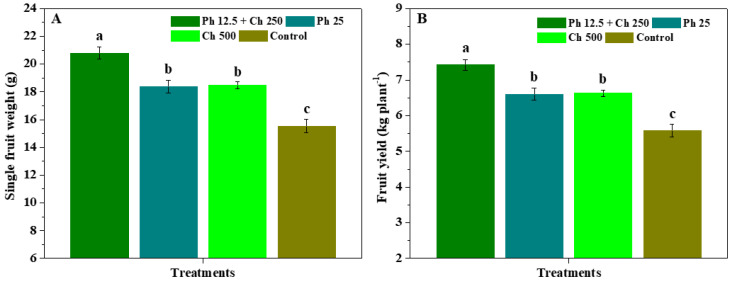
The impacts of physcion + chitosan, physcion, and chitosan on the weight (**A**) and yield (**B**) of fruits. The significant differences in the five treatments at 5% (*p* < 0.05) level are stood by different letters. Error bar represents the SD of three duplicates.

**Table 1 antibiotics-11-01661-t001:** The climatic and nutritional information of *R. roxburghii* garden.

Indices	Amount	Indices	Amount	Indices	Amount
Average altitude	1384 m	Organic matter	13.65 g kg^−1^	Available potassium	28.23 mg kg^−1^
Annual sunshine	1265 h	Total nitrogen	1.41 g kg^−1^	Exchangeable calcium	18.16 cmol kg^−1^
Frostless season	280 days	Total phosphorus	1.70 g kg^−1^	Exchangeable magnesium	309.24 mg kg^−1^
Average temperature	13.9 °C	Total potassium	1.22 g kg^−1^	Available zinc	0.66 mg kg^−1^
Annual rainfall	1100 mm	Available nitrogen	57.18 mg kg^−1^	Available iron	6.62 mg kg^−1^
pH	6.42	Available phosphorus	4.78 mg kg^−1^	Available boron	0.16 mg kg^−1^

**Table 2 antibiotics-11-01661-t002:** The efficacy of physcion + chitosan, physcion, and chitosan control powdery mildew.

Treatments	Disease Index before Spraying Fungicides	After Spraying Fungicides			
		Disease Index of 7 d	Control Effect (%) of 7 d	Disease Index of 14 d	Control Effect (%) of 14 d
Ph 12.5 + Ch 250	2.27 ± 0.04 ^a^	2.83 ± 0.18 ^d^	92.65 ± 1.58 ^a^	3.47 ± 0.04 ^d^	90.68 ± 0.10 ^a^
Ph 25	2.23 ± 0.04 ^a^	3.47 ± 0.09 ^c^	83.62 ± 2.42 ^b^	4.79 ± 0.17 ^c^	80.43 ± 1.02 ^b^
Ch 500	2.24 ± 0.06 ^a^	4.46 ± 0.11 ^b^	70.75 ± 2.96 ^c^	5.15 ± 0.14 ^b^	77.80 ± 1.11 ^c^
Control	2.28 ± 0.03 ^a^	9.92 ± 0.66 ^a^	-	15.38 ± 0.13 ^a^	-

The mean ± SD of three duplicates indicates value. The significant differences in the five treatments at 5% (*p* < 0.05) level are stood by different letters.

**Table 3 antibiotics-11-01661-t003:** The impacts of physcion + chitosan, physcion, and chitosan on fruit quality.

Treatments	Ascorbic Acid (mg g^−1^)	Soluble Protein (%)	Soluble Sugar (%)	Total Acidity (%)	Soluble Solids (%)	Flavonoids (mg·g^−1^)	Triterpenes (mg·g^−1^)	SOD Activity (U g^−1^ FW)
Ph 12.5 + Ch 250	23.42 ± 0.95 ^a^	15.91 ± 0.49 ^a^	4.24 ± 0.10 ^a^	1.53 ± 0.05 ^a^	12.53 ± 0.15 ^a^	6.35 ± 0.23 ^a^	20.74 ± 0.29 ^a^	714.86 ± 16.47 ^a^
Ph 25	20.89 ± 0.74 ^b^	14.54 ± 0.36 ^b^	3.75 ± 0.09 ^b^	1.41 ± 0.06 ^b^	11.65 ± 0.32 ^b^	5.77 ± 0.12 ^b^	17.86 ± 0.26 ^b^	654.31 ± 16.08 ^b^
Ch 500	21.56 ± 0.96 ^b^	14.78 ± 0.62 ^b^	3.89 ± 0.10 ^b^	1.45 ± 0.08 ^b^	11.86 ± 0.20 ^b^	5.94 ± 0.27 ^b^	18.72 ± 0.56 ^b^	678.45 ± 19.95 ^b^
Control	18.11 ± 0.65 ^c^	13.53 ± 0.13 ^c^	3.13 ± 0.11 ^c^	1.22 ± 0.07 ^b^	10.65 ± 0.34 ^c^	5.18 ± 0.12 ^c^	15.18 ± 0.56 ^c^	558.59 ± 28.06 ^c^

The mean ± SD of three duplicates indicates value. The significant differences in the five treatments at 5% (*p* < 0.05) level are stood by different letters.

**Table 4 antibiotics-11-01661-t004:** The impacts of physcion + chitosan, physcion, and chitosan on fruit amino acids.

Treatments	EAA (mg kg^−1^)	NAA (mg kg^−1^)	TAA (mg kg^−1^)	The Proportion of EAA in TAA (%)	EAA/NAA
Ph 12.5 + Ch 250	88.32 ± 2.82 ^a^	339.09 ± 11.28 ^a^	468.27 ± 36.39 ^a^	18.91 ± 0.90 ^a^	0.2607 ± 0.0127 ^a^
Ph 25	73.29 ± 3.65 ^b^	315.77 ± 12.71 ^ab^	411.25 ± 12.19 ^b^	17.85 ± 1.40 ^ab^	0.2323 ± 0.0133 ^bc^
Ch 500	78.63 ± 1.54 ^b^	323.36 ± 10.53 ^a^	425.75 ± 15.47 ^ab^	18.48 ± 0.60 ^a^	0.2432 ± 0.0037 ^ab^
Control	64.33 ± 5.96 ^c^	296.77 ± 15.80 ^b^	384.65 ± 31.57 ^b^	16.71 ± 0.25 ^b^	0.2168 ± 0.0184 ^c^

The mean ± SD of three duplicates indicates value. The significant differences in the five treatments at 5% (*p* < 0.05) level are stood by different letters. EAA = essential amino acids, NAA = nonessential amino acids, and TAA = total amino acids.

## Data Availability

The datasets used or analyzed during the current study available from the corresponding author upon reasonable request.

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
