# Peer review of "Association of Physcion and Chitosan Can Efficiently Control Powdery Mildew in Rosa roxburghii"

_antibiotics, 2022, doi:10.3390/antibiotics11111661_

Round 1
Reviewer 1 Report
Dear Authors.
Below are the comments/observations/changes you should make to your manuscript. You must take everything into account in order to update the manuscript.
1) Include the impact in economic figures or quantitative indicators on the impact/damage generated by Powdery Mildew on other plant species and on Rosa roxburghii.
2) All of section 2 should be rewritten. You must write in detail all materials and methods performed/used in this work. Remember to include references. You must include all references that support/validate what has been done.
3) You should include in the manuscript why and for what purpose you evaluated only the treatments Ph 12.5 + Ch 250, Ph 25, and Ch 500, and not others.
4) In section 4, include a paragraph on the strengths and weaknesses of your work.
Author's update the manuscript
Best regards
Reviewer
Author Response
1st Reviewer
Comment 1: Below are the comments/observations/changes you should make to your manuscript. You must take everything into account in order to update the manuscript.
Response: We sincerely thank the reviewer for the positive comments, careful reviews and warm work to our work! We also sincerely thank you for your hard corrections on our manuscript! Reviewers' comments are extremely constructive and valuable, and very helpful for revising and improving our manuscript, as well as the important guiding significance to our researches. We have studied carefully the reviewers' comments and made substantial revisions which we sincerely hope meet with approval. The responds to the reviewers' comments and the corrections in the revised manuscript are as flows. Thank you most sincerely!
Comment 2: 1) Include the impact in economic figures or quantitative indicators on the impact/damage generated by Powdery Mildew on other plant species and on Rosa roxburghii.
Response: Thanks very much for the reviewer's careful reviews on our manuscript! The reviewers' comment is extremely constructive. The corresponding statement has been added and revised as "Powdery mildew is an extremely serious disease of all Rosa roxburghii production regions in China and frequently causes 30~40% of economic losses." and " Powdery mildew caused by Sphaerotheca pannosa (Wallr.) Lev is an extremely serious fungal disease of all R. roxburghii production regions in China, which has been responsible for seriously growth restrictions and quality declines, as well as 30~40% of yield or economic losses [7–9]." which marked in blue in the revised manuscript. Thank you most sincerely! (See lines 9-10, 35-39)
Comment 3: 2) All of section 2 should be rewritten. You must write in detail all materials and methods performed/used in this work. Remember to include references. You must include all references that support/validate what has been done.
Response: We sincerely thank the reviewer for the good comments, careful review and warm work to our manuscript! The corresponding statement has been added and revised as "In our recently report, we found that 500 mg L-1 chitosan had a good control effect on R. roxburghii powdery mildew with the efficacy of 77.80% after 14 d of application [9]. Moreover, Zhang et al. [34] found that 12.5~25.0 mg L-1 0.5% physcion AS had a significant inhibitory effect on R. roxburghii powdery mildew with the control efficacy of 77.01%~82.26% after 14 d of application. Considering the possible synergistic interaction between chitosan and physcion, the reduced application of fungicides and the economic cost of field control experiment, thus three natural fungicide groups and one control group were devised to control powdery mildew. They were: (1) 12.5 mg L-1 0.5% physcion AS + 250 mg L-1 chitosan (Ph 12.5 + Ch 250), (2) 25 mg L-1 0.5% physcion AS (Ph 150), (3) 500 mg L-1 chitosan (Ch 500), (4) irrigation water (as control). The foliar spray method was adopted for spraying fungicide liquid. Three replicates…." and " The photosynthetic rate (Pn), transpiration rate (Tr), and water use efficiency (WUE) of R. roxburghii leaves were surveyed by a portable LI-6400XT photosynthesis measurement (LI-COR Inc., Lincoln, NE, USA) on May 12 in 2022 [35]. Subsequently, R. roxburghii leaves from five orientations on each tree were stochastically gathered for determining their chlorophyll content and disease resistance parameters. Chlorophyll content with an ethanol/acetone (v/v, 1:2) extraction was measured using an UV-5800PC spectrophotometer [9]. Folin-ciocalteu, HCl–methyl alcohol, coomassie brilliant blue, anthrone colorimetric, ninhydrin colorimetry, and thiobarbituric acid methods were used for determining the phenolics, flavonoids, protein, sugar, proline (Pro), and malonaldehyde (MDA) contents of leaves, respectively [9,13,36]. Moreover, the SOD and polyphenoloxidase (PPO) activities of leaves were determined by the nitrogen blue tetrazole and catechol methods, respectively [9]. R. roxburghii fruits from five orientations on each tree were stochastically harvested on August 31 in 2022. The weight and yield of fruits were measured based on the method of Li et al [7, 19]. The ascorbic acid, sugar, solid, protein, acidity, flavonoids, triterpenes, and SOD activity of fruits were determined as the methods of Wang et al [37] and Zhang et al [9, 36]. High performance liquid chromatography (HPLC, ThermoFisher U3000, Waltham, MA, USA) method was used for checking ascorbic acid. Soluble solid was determined by a digital refractometer (Deke Machinery Technology Co. Ltd., Hebei, China). Titratable acidity and triterpenes were measured by the NaOH titration and vanillin-glacial acetic acid colorimetric spectrophotometry methods, respectively. The determination methods of sugar, protein, flavonoids, and SOD activity were the same as above. Additionally, hydrolyzed amino acids of fruits were measured by a HPLC system based on the method of Zhang et al [35]. Subsequently, essential, nonessential, and total amino acids, as well as essential amino acids / nonessential amino acids and the proportion of essential amino acids in total amino acids were calculated." which marked in blue in the revised manuscript. Thank you most sincerely! (See lines 101-111, 123-148)
Comment 4: 3) You should include in the manuscript why and for what purpose you evaluated only the treatments Ph 12.5 + Ch 250, Ph 25, and Ch 500, and not others.
Response: We sincerely thank the reviewer for the careful reviews! The reviewers' comment is extremely valuable. The corresponding statement has been added and revised as "In our recently report, we found that 500 mg L-1 chitosan had a good control effect on R. roxburghii powdery mildew with the efficacy of 77.80% after 14 d of application [9]. Moreover, Zhang et al. [34] found that 12.5~25.0 mg L-1 0.5% physcion AS had a significant inhibitory effect on R. roxburghii powdery mildew with the control efficacy of 77.01%~82.26% after 14 d of application. Considering the possible synergistic interaction between chitosan and physcion, the reduced application of fungicides and the economic cost of field control experiment, thus three natural fungicide groups and one control group were devised to control powdery mildew. They were: (1) 12.5 mg L-1 0.5% physcion AS + 250 mg L-1 chitosan (Ph 12.5 + Ch 250), (2) 25 mg L-1 0.5% physcion AS (Ph 150), (3) 500 mg L-1 chitosan (Ch 500), (4) irrigation water (as control). The foliar spray method was adopted for spraying fungicide liquid. Three replicates…." which marked in blue in the revised manuscript. We sincerely hope to get your understanding and recognition. Thank you most sincerely! (See lines 101-111)
Comment 5: 4) In section 4, include a paragraph on the strengths and weaknesses of your work. Author's update the manuscript
Response: We sincerely thank the reviewer for the careful reviews! The reviewers' comment is extremely valuable. The corresponding statement has been added and revised as " Chemical fungicide residuals will inevitably pose a potential threat for the ecosystem and human beings [12–14]. Currently, natural products with non-toxicity, high efficacy and low risks as an effective alternative to chemical fungicides, are increasingly used in agriculture production and also growingly preferred by consumers [15–17]. Physcion, a natural anthraquinone derivant, has been demonstrated to possess potential anti-neoplastic and anti-microbial characteristics and widely used in pharmaceutical and agriculture fields [26–34, 42-43]. Simultaneously, chitosan is a natural, nontoxic, biodegradable, and renewable product extensively used in food, cosmetics, and agriculture fields [22–24]. In this study, the association of physcion and chitosan could efficiently control R. roxburghii powdery mildew with the efficacy of 90.68%~92.65%, as well as effectively ameliorate the resistant, photosynthesis, yield and quality of R. roxburghii. Moreover, 141 days (From April 12 to August 31) of the safe interval time for fruits was immensely long. Consequently, the possible safe risks caused by physcion or chitosan are also practically nonexistent. The findings emphasize that 12.5 mg L-1 0.5% physcion AS + 250 mg L-1 chitosan can be nominated as a can-didate association for controlling R. roxburghii powdery mildew and ameliorating its resistance, growth, and quality. However, the interaction mechanism of physcion and chitosan on the pathogen of powdery mildew should be further revealed in the future research, so as to fully understand the control mechanism of this association on R. roxburghii powdery mildew." which marked in blue in the revised manuscript. Thank you most sincerely! (See lines 335-353)
We deeply appreciate for reviewer’s warm work earnestly, and hope that the correction will meet with approval. Once again, thank you very much for your comments and suggestions.
Reviewer 2 Report
Success !

Author Response
2nd Reviewer
Comment 1: What is the main question addressed by the research? Is it relevant andinteresting?
The study concerns the efficacy of the combination of physcion and chitosancompounds, as well as of the two compounds taken separately, on diseaseresistance, growth, production and fruit quality in the species Rosa roxburghii. The subject is of interest, the effect of the experimented Ph+Ch combinationbeing one of high efficiency in combating powdery mildew, but also improvingthe nutritional value of R. roxburghii fruits.
How original is the topic? What does it add to the subject area compared withother published material?
Research on the tested compounds was also carriedout previously, mainly on chitosan, but the article has an original approach bystudying the physcion and chitosan combination, and the use of physcion in thetreatment of the powdery mildew at Rosa roxburgii species. These are in factthe elements of novelty compared to other research.
Is the paper well written? Is the text clear and easy to read?
The article iswell structured, generally clear and easy to read. The research uses a preciseprotocol. However, in some places, the terminology is wrong, the titles of thetables are poorly formulated, there are inappropriate numbering andcumbersome text forms.These does not affect seriously the quality of thearticle, but they need to be reviewed and corrected, including by a native English speaker.
Are the conclusions consistent with the evidence and arguments presented?Do they address the main question posed?
The conclusions cover the aspects studied, are consistent with the results and arguments but they are too brief. They need to be better developed and with a formulation in the tone of the high scientific value of the article.
Response: We sincerely thank the reviewer for the positive comments, careful reviews and warm work to our work! Reviewers' comments are extremely constructive and valuable, and very helpful for revising and improving our manuscript, as well as the important guiding significance to our researches. We try our best to improve the English language, and the manuscript also was polished by Doctor Kashif Ali Solangi. Moreover, the conclusions has been further revised and developed. We have also studied carefully the reviewers' comments and have substantial revisions which we sincerely hope meet with approval. The responds to the reviewers' comments and the corrections in the revised manuscript are as flows. Thank you most sincerely! (See lines 358-363)
Comment 2: Line 9-12: review the text in English. Some terms are inappropriate, for example manifest on the row 12, which need to be replaced by show, reveal etc.
Response: Special thanks to you for your good comments and careful reviews! The corresponding statement has been revised as "Rosa roxburghii powdery mildew is an extremely serious disease of all production regions in China. Natural products are considered as the excellent alternatives to chemical fungicides. In this work, we investigated the efficacy of physcion used together with chitosan controls R. roxburghii powdery mildew and impacts its resistant, growth, yield, and quality. The results reveal…" which marked in blue in the revised manuscript. Thank you most sincerely! (See lines 9-13)
Comment 3: Line 33: grown instead of cultured and cultivation area instead of culturing proportion.
Response: We sincerely thank the reviewer for the careful reviews and good comment! "grown" and "culturing proportion" have respectively been revised as "cultured" and "cultivation area" which marked in blue in the revised manuscript. Thank you most sincerely! (See line 34)
Comment 4: Line 40: may be local research instead of academics...
Response: We sincerely thank the reviewer for the careful reviews and good comment! "academics" has been revised as "research" which marked in blue in the revised manuscript. Thank you most sincerely! (See line 41)
Comment 5: Line 88-92: the presentation of the company producing the products seems aforced matter...
Response: Special thanks to you for your good comments! The corresponding presentations have been added and revised which marked in blue in the revised manuscript. Thank you most sincerely! (See lines 88-92)
Comment 6: Table 1: separate the columns.
Response: Thanks very much for the reviewer's good advice to our manuscript! The corresponding columns have been separated which marked in blue in the revised manuscript. Thank you most sincerely! (See Table 1)
Comment 7: Line 101: wrong wording, incomprehensible, of the first sentence.
Response: Thanks very much for the reviewer's careful reviews and good comment on our manuscript! The corresponding statement has been revised as "The foliar spray method was adopted for spraying fungicide liquid." which marked in blue in the revised manuscript. Thank you most sincerely! (See lines 110-111)
Comment 8: Line 104: may be replicates, not duplicates ...
Response: We sincerely thank the reviewer for the careful reviews and warm work to our work! "duplicates" has been revised as "replicates" which marked in blue in the revised manuscript. Thank you most sincerely! (See line 111)
Comment 9: Line 128: eliminate simultaneously, from text.
Response: Thanks very much for the reviewer's careful reviews on our manuscript! "simultaneously" has been deleted. Thank you most sincerely!
Comment 10: Line 137: check the wards.
Response: Special thanks to you for your good comments and careful reviews! "sysrem" has been revised as "system" which marked in blue in the revised manuscript. Thank you most sincerely! (See line 152)
Comment 11: Line 142...162 and throughout the text: check the use of the term dramatically in the interpretation of the results. It is inappropriate in most contexts and needs to be replaced.
Response: We sincerely thank the reviewer for the careful reviews and good comment! "dramatically" has been replaced which marked in blue throughout the revised manuscript. Thank you most sincerely! (See lines 157, 160, 165, 177, 191, 208-210, 237)
Comment 12: Line 156: You're talking about figure 2, but it's actually figure 1. Correct. Also here, you talk about pyraclostrobin and it is missing from the charts. Correct it. Line 159-164: coherently reformulates the text. It's hard to understand.
Response: We sincerely thank the reviewer for the careful reviews and good comment! We are so sorry for our carelessness. The corresponding statement has been revised as "Figure 1 shows the impacts of physcion + chitosan, physcion, and chitosan on the resistant substances of R. roxburghii leaves. In comparison to control, Ph 12.5 + Ch 250, Ph 25, and Ch 500 significantly (p < 0.05) heightened the flavonoids, protein, and sugar contents of leaves, and Ph 12.5 + Ch 250 and Ch 500 significantly (p < 0.05) promoted their phenolics. Meanwhile, the flavonoids, protein, and sugar of leaves controlled by Ph 12.5 + Ch 250 were markedly (p < 0.05) greater than those of Ph 25 or Ch 500. The phenolics of R. roxburghii leaves controlled by Ph 12.5 + Ch 250 was also observably (p < 0.05) greater than that of Ph 25, but was not significantly (p < 0.05) different with that of Ch 500. Moreover, the phenolics, flavonoids, protein, and sugar of leaves were no substantially (p < 0.05) differences in treatments of Ph 25 and Ch 500, and those of Ch 500 were slightly exceeded to those of Ph 25." which marked in blue in the revised manuscript. Thank you most sincerely! (See lines 171, 174-181)
Comment 13: Line 171(Figure 1) and in all the others: you present all the experiment alvariants, but the title refers only to the Ph+Ch variant. Correct, with reference to all the combinations experienced.
Response: Special thanks to you for your good comments! All titles of Figure 1~4 and Table 2~4 have been added all the combinations experienced which marked in blue in the revised manuscript. Thank you most sincerely! (See lines 167, 186, 202, 218, 242, 244, 256)
Comment 14: Line 201: replace declare that with shows that, or reveal that.
Response: Thanks very much for the reviewer's good advice to our manuscript! "declare" has been revised "reveal" which marked in blue in the revised manuscript. Thank you most sincerely! (See line 214)
Comment 15: Line 216: replace encouraged with stimulates and boosted with increased.
Response: Thanks very much for the reviewer's careful reviews and good comment on our manuscript! "encouraged" and "boosted" have respectively been revised as "stimulated" and "increased" which marked in blue in the revised manuscript. Thank you most sincerely! (See line 229, 232)
Comment 16: Line 243-244: explain below the table, in a legend, with an asterisk, the acronyms EAA, NAA, TAA (ex. EAA=essential amino ac..)
Response: We sincerely thank the reviewer for the careful reviews and warm work to our work! The corresponding explain has been added and revised as "EAA=essential amino acids, NAA=nonessential amino acids, and TAA=total amino acids." which marked in blue in the revised manuscript. Thank you most sincerely! (See line 258)
Comment 17: Line247: replace documents with research and add the scientific name of the disease in parentheses.
Response: Thanks very much for the reviewer's careful reviews on our manuscript! The corresponding statement has been added and revised as "Many researches have evidenced that physcion could effectively prevent and control the powdery mildew (Blumeria graminis, Podosphaera xanthii, Sphaerotheca fuliginea, etc.), gray mold (Botrytis cinerea), and downy mildew (Pseudoperonospora cubensis) of various plants [26–33]." which marked in blue in the revised manuscript. Thank you most sincerely! (See lines 260-263)
Comment 18: Line 263: relace manifest by shows...
Response: Thanks very much for the reviewer's careful reviews on our manuscript! The corresponding statement has been revised as "show" which marked in blue in the revised manuscript. Thank you most sincerely! (See line 277)
Comment 19: Line 272:...while PPO..
Response: Special thanks to you for your good comments and careful reviews! The corresponding statement has been revised as "while PPO" which marked in blue in the revised manuscript. Thank you most sincerely! (See line 286)
Comment 20: Line 278 and 293: check words.
Response: We sincerely thank the reviewer for the careful reviews and good comment! The corresponding words have been revised which marked in blue in the revised manuscript. Thank you most sincerely! (See lines 292, 307)
Comment 21: Line 302: replace encourage by stimulates.
Response: We sincerely thank the reviewer for the careful reviews and good comment! The corresponding words have been revised as "stimulated" which marked in blue in the revised manuscript. Thank you most sincerely! (See line 316)
Comment 22: Line 306:check astricted... may be affected..
Response: Special thanks to you for your good comments! The corresponding words have been revised as "affected" which marked in blue in the revised manuscript. Thank you most sincerely! (See line 320)
Comment 23: Line 328: develop a little the sentence related to the 141 days, because it is not clear.
Response: Thanks very much for the reviewer's good advice to our manuscript! The corresponding words have been revised as "Moreover, 141 days (From April 12 to August 31) of the safe interval time for fruits was immensely long." which marked in blue in the revised manuscript. Thank you most sincerely! (See lines 346-347)
We appreciate for reviewer’s warm work earnestly, and hope that the correction will meet with approval. Once again, thank you very much for your comments and suggestions.
Round 2
Reviewer 1 Report
Dear Authors
I have reviewed your update report and the updated manuscript, and I have the following observations which should be made in full.
1) The methodology as well as the discussion are vitally important sections of an article. The central purpose of an article is to contribute and transfer knowledge to the globalised world of innovation, Science and Technology.
2) It also aims to disseminate in a "clear and precise manner" the results of research carried out in a particular area of knowledge. It can also encourage the development of innovative experimental methods.
3) When reviewing the update reports and the manuscript, not all references listed were found. References listed in the report [7], [9], [13], [19], [36] and [36], are not found in their entirety in the manuscript.
4) When you trace these references, i.e. review, consult, read and analyse them, you find that they do not provide concrete, clear, precise, detailed information on the methodologies you have used to validate the section of the results.
5) Therefore the results section of your manuscript are not valid, because the methodology for each of the analytical parameters is not detailed, and the supporting references are equally inconsistent (no detailed information on how the analytical determinations were performed).
6) Authors, remember that your work can be cited by other authors who can replicate your methodology in other fields of research.
7) You must then rewrite in detail each of the analytical methods used in your work.
Authors, please make the requested update to your manuscript. The review process of your manuscript is almost over.
Best regards
Reviewer
Author Response
Comment 1: I have reviewed your update report and the updated manuscript, and I have the following observations which should be made in full.
Response: We sincerely thank the reviewer for the positive comments, careful reviews and warm work to our work! Reviewers' comments are extremely constructive and valuable, and very helpful for revising and improving our manuscript, as well as the important guiding significance to our researches. We have studied carefully the reviewers' comments and made substantial revisions which we sincerely hope meet with approval. The responds to the reviewers' comments and the corrections in the revised manuscript are as flows. Thank you most sincerely!
Comment 2: 1) The methodology as well as the discussion are vitally important sections of an article. The central purpose of an article is to contribute and transfer knowledge to the globalised world of innovation, Science and Technology.
Response: Thanks very much for the reviewer's careful reviews on our manuscript! Reviewers' comments are extremely teaching meaning, and the important guiding significance to our researches. We have studied carefully the reviewers' comments and made substantial revisions which we sincerely hope meet with approval. Thank you most sincerely!
Comment 3: 2) It also aims to disseminate in a "clear and precise manner" the results of research carried out in a particular area of knowledge. It can also encourage the development of innovative experimental methods.
Response: We sincerely thank the reviewer for the good comment and warm work to our manuscript! Thank you for your hard teaching. We tried our best to revise the experimental method of the article in detail. We sincerely hope to get your understanding and recognition. Thank you most sincerely! (See lines 123-206, 352, 373, 399, 509-516)
Comment 4: 3) When reviewing the update reports and the manuscript, not all references listed were found. References listed in the report [7], [9], [13], [19], [36] and [36], are not found in their entirety in the manuscript.
Response: We sincerely thank the reviewer for the careful reviews! I apologize for our misquote. We tried our best to revise the experimental method of the article in detail. We sincerely hope to get your understanding and recognition. Thank you most sincerely! (See lines 123-206, 352, 373, 399, 509-516)
Comment 5: 4) When you trace these references, i.e. review, consult, read and analyse them, you find that they do not provide concrete, clear, precise, detailed information on the methodologies you have used to validate the section of the results.
Response: We sincerely thank the reviewer for the careful reviews! Thank you for your hard teaching and criticism. We tried our best to revise the experimental method of the article in detail. We sincerely hope to get your understanding and recognition. Thank you most sincerely! (See lines 123-206, 352, 373, 399, 509-516)
Comment 6: 5) Therefore the results section of your manuscript are not valid, because the methodology for each of the analytical parameters is not detailed, and the supporting references are equally inconsistent (no detailed information on how the analytical determinations were performed).
Response: Thanks very much for the reviewer's good advice to our manuscript! The corresponding experimental methods have been separated which marked in blue in the revised manuscript. Thank you most sincerely! (See lines 123-206, 352, 373, 399, 509-516)
Comment 7: 6) Authors, remember that your work can be cited by other authors who can replicate your methodology in other fields of research.
Response: Thanks very much for the reviewer's careful reviews and good comment on our manuscript! The corresponding statement has been revised as " The photosynthetic rate (Pn), transpiration rate (Tr), and water use efficiency (WUE) of R. roxburghii leaves were surveyed by a portable LI-6400XT photosynthesis meas-urement (LI-COR Inc., Lincoln, NE, USA) on May 12 in 2022 [35]. Subsequently, R. rox-burghii leaves from five orientations on each tree were stochastically gathered for de-termining their chlorophyll content and disease resistance parameters. Chlorophyll content with an ethanol/acetone (v/v, 1:2) extraction was measured using an UV-5800PC spectrophotometer [9]. The method of Cao et al [36] was used for deter-mining the phenolics and flavonoids of leaves, 2.00 g of sample was ground in 20 mL of HCl-methyl alcohol (1%, v/v) and centrifuged (8,000× g, 8 min, 4℃) for extracting 1 h without light, then the supernatant was checked at OD 280 nm and OD 325 nm, re-spectively. Coomassie brilliant blue method was used for determining leaf protein content, 2.00 g of sample was added with 14 mL of distilled water, extracted by ultra-sonic, centrifuged, and the result was measured by the mass concentration of bovine serum albumin [36]. Anthrone colorimetric method was used for determining the sug-ar content of leaves [36]. Ninhydrin colorimetry method was used for determining the proline (Pro) content of leaves, 0.50 g of sample was added into 5 mL of 3% sulfosali-cylic acid solution to extract in boiling water bath for 10 min, then cooled and filtered to prepare the extract solution; 2 mL of extraction solution was added into 2 mL of gla-cial acetic acid and 2 mL of acid ninhydrin reagent, heated in boiling water bath for 30min, and the solution turned red; After cooling, it was added 4 mL of toluene to ex-tract for about 20 seconds with a quick mixer, and centrifuged for 5 min (3000 r); The upper liquid was measured at OD 520 nm, and toluene as the control [36]. Thiobarbitu-ric acid method was used for determining the malonaldehyde (MDA) content of leaves, 1.00 g of sample was added into 5 mL of 10% trichloroacetic acid ice bath to grind into homogenate, centrifuged for 10 min (4℃, 4000 r), and the supernatant was used as the extract; 2 mL of extract was added into 2 mL of 0.6% thiobarbituric acid solution, boiled in a boiling water bath for 15 min, and cooled and centrifugated, and then measured at OD 450 nm and OD 532 nm, 2 mL of 10% trichloroacetic acid was used as the control [36]. The nitrogen blue tetrazole method was used for determining the SOD activity of leaves, 0.50 g of sample was added into 1 mL of 0.5 mol/L phosphoric acid buffer (pH=7.8) and ground into homogenate in ice bath, centrifuged for 20 min (4 ℃, 4000 r); The 3 mL reaction system contained 0.3 mL of 750 umol/L azobenzene tetrazole solu-tion, 0.05 mL of extract crude enzyme solution, 0.3 mL of 20 umol/L riboflavin solution, 0.3 mL of 130 mmol/L methionine solution, 0.3 mL of 100 umol/L EDTA-Na2, 1.5 mL of 0.05 mol/L phosphate buffer solution and 0.25 mL of distilled water; After mixing, one branch pipe was put in a dark place, and the other pipes reacted for 20 min under 4000 Lx light; After the reaction, the OD 560 nm was measured with a non-illuminated tube as the control, and one enzyme activity unit was 50% inhibition of NBT photoreduction [36]. The catechol method was used for determining the polyphenoloxidase (PPO) ac-tivity of leaves, 0.50 g of sample was added with 0.05 g of polyvinylpyrrolidone and 2 mL of 0.1 mol/L phosphate buffer (pH=6.5) to grind into a homogenate in an ice bath; The volume was fixed to 5 mL, filtered with nylon cloth, centrifuged for 15 min (4℃, 8000 r), and the supernatant was the crude enzyme extract; 0.2 mL of enzyme extract was added into 2.8 mL of phosphate buffer containing 0.02 mol/L catechol (0.1 mol/L, pH 6.8) to mix; After reacting in water bath at 30 ℃ for 2 min, OD 398 nm was deter-mined with the same volume of extract as the control, the change of OD 398 nm value of 0.01 per minute was taken as one enzyme activity unit [36].
- roxburghii fruits from five orientations on each tree were stochastically har-vested on August 31 in 2022. The weight and yield of fruits were measured based on the method of Li et al [7, 19]. The ascorbic acid, sugar, solid, protein, acidity, flavonoids, triterpenes, and SOD activity of fruits were determined as the methods of Cao et al [36]. High performance liquid chromatography (HPLC, ThermoFisher U3000, Waltham, MA, USA) method was used for checking ascorbic acid, 0.25 g sample was added into 1.5 mL of 0.3% metaphosphoric acid solution, and then frozen, ground and mixed for 2 min, centrifuged at 4 ℃ for 15 min (5000 r), and the supernatant was put through 0.22μm water phase filtration membrane to be measured; The chromatographic conditions were as follows: ZORBAX SB-C18 column (250 mm×4.6 mm, 5μm), 30℃ of column temperature was, diode array detector, 254 nm of the detection wavelength, 0.1 mol/L disodium phosphate solution (pH 2.7) of mobile phase, 1 mL/min of flow rate, 10 μL of injection volume. Soluble solid was determined by a digital refractometer (Deke Ma-chinery Technology Co. Ltd., Hebei, China). Titratable acidity of fruits was measured by the NaOH titration method [36]. Triterpenes of fruits was measured by the vanil-lin-glacial acetic acid colorimetric spectrophotometry method, 1.00 g of sample was added into 50 mL absolute ethanol to ultrasonic extract for 1 h (50℃), the supernatant was centrifuged, and then measured at OD 548 nm, its result was calculated by the mass concentration of ursolic acid [36]. The determination methods of sugar, protein, flavo-noids, and SOD activity were the same as above. Additionally, hydrolyzed amino acids of fruits were measured by a HPLC system based on the method of Zhang et al [37]. 0.10 g of sample was added into a 20 mL hydrolysis tube and 16 mL of 6 mol/L HCl so-lution to vacuum degas for 30 min, filled nitrogen to seal the tube, hydrolyzed at 110℃ for 22~24 hours and cooled down, and then transfered into a 50 mL volumetric flask for constant volume by the deionized water; 1 mL of hydrolysate was deacidified and drain under vacuum, and then added 1 mL of 0.02 mol/L HCl solution to fully dissolve it; 500 μ L above solution was added 250 μL of 1 mol/L triethylamine acetonitrile solution and add 25 μL of 0.1 mol/L phenyl isothiocyanate acetonitrile solution, leaved it at room temperature for 1h, then added 2 mL n-hexane to shake it violently, and leaved it to stand for 10min, finally the lower solution was put through 0.22 μM aqueous phase membrane for analysis. Chromatographic conditions: 0.1 mol/L sodium ace-tate-acetonitrile solution (93:7, v/v) of mobile phase A, acetonitrile-water (8:2, v/v) of mobile phase B, 1.0 mL/min of flow rate, 10 μL of injection volume, 254 nm of wave-length, and 40℃ of column temperature, as well as the column filler was octadecyl silane bonded silica gel (4.6 mm × 250 mm,5 μ m). Subsequently, essential, nonessential, and total amino acids, as well as essential amino acids / nonessential amino acids and the proportion of essential amino acids in total amino acids were calculated." which marked in blue in the revised manuscript. Thank you most sincerely! (See lines 123-206, 352, 373, 399, 509-516)
Comment 8: 7) You must then rewrite in detail each of the analytical methods used in your work.
Response: We sincerely thank the reviewer for the careful reviews and warm work to our work! Thank you again for your criticism and instruction. We tried our best to revise the experimental method of the article in detail. We sincerely hope to get your understanding and recognition. Thank you most sincerely! (See lines 123-206, 352, 373, 399, 509-516)
We appreciate for reviewer’s warm work earnestly, and hope that the correction will meet with approval. Once again, thank you very much for your comments and suggestions.